# Multimodal Query Suggestion with Multi-Agent Reinforcement Learning from Human Feedback

## ABSTRACT

In the rapidly evolving landscape of information retrieval, search engines strive to provide more personalized and relevant results to users. Query suggestion systems play a crucial role in achieving this goal by assisting users in formulating effective queries. However, existing query suggestion systems mainly rely on textual inputs, potentially limiting user search experiences for querying images. In this paper, we introduce a novel Multimodal Query Suggestion (MMQS) task, which aims to generate query suggestions based on user query images to improve the intentionality and diversity of search results. We present the RL4Sugg framework, leveraging the power of Large Language Models (LLMs) with Multi-Agent Reinforcement Learning from Human Feedback to optimize the generation process. Through comprehensive experiments, we validate the effectiveness of RL4Sugg, demonstrating a 18% improvement compared to the best existing approach. Moreover, the MMQS has been transferred into real-world search engine products, which yield enhanced user engagement. Our research advances query suggestion systems and provides a new perspective on multimodal information retrieval.

## CCS CONCEPTS

• **Information systems** → *Multimedia information systems*.

## KEYWORDS

multimodal query suggestion, multi-agent reinforcement learning from human feedback, vision-language pre-training

**ACM Reference Format:**
Anonymous Author(s). 2024. Multimodal Query Suggestion with Multi-Agent Reinforcement Learning from Human Feedback. In *Proceedings of the ACM Web Conference 2024 (WWW'24), May 13–17, 2024, Singapore.* ACM, New York, NY, USA, 12 pages. https://doi.org/10.1145/3543507.3583304

## 1 INTRODUCTION

Search engines have become an indispensable tool for information retrieval, aiding users in finding relevant content in vast online repositories. Traditional keyword-based search methods [23, 46], while effective, often require users to precisely articulate their information needs, leading to potential challenges in formulating accurate queries. To enhance the search experience and provide more user-friendly alternatives, query suggestion systems have

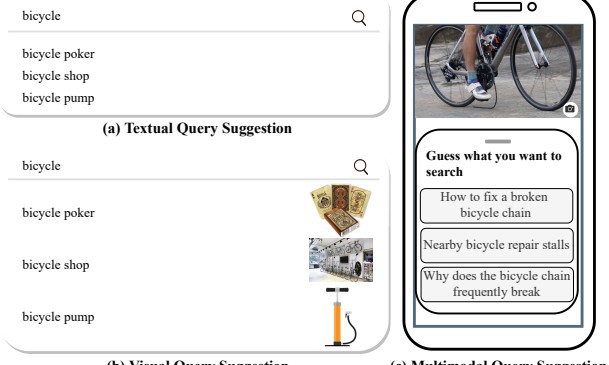

**Figure 1: Illustration of MMQS problem.**

gained prominence. These systems aim to generate relevant and contextually appropriate suggestions based on users' current query input, reducing the cognitive burden on users and increasing the efficiency of information discovery.

There are two well-established query suggestion systems that have been extensively studied: Textual Query Suggestion (TQS) [2, 5, 14, 15, 17, 39] and Visual Query Suggestion (VQS) [28, 49–51]. In TQS, it is capable to automatically suggest a list of keywords based on users' current queries, a feature that many existing search engines have already implemented. Its primary purpose is to assist users in formulating their search intents clearly and conveniently (as illustrated in Figure 1(a)). In VQS, the suggestions generated by TQS might be inadequate for users who lack familiarity with the suggested terms. To address this issue, incorporating visual examples along with the suggestions can greatly improve the user experience and help users better understand the context (as illustrated in Figure 1(b)). The limitation of these systems is that they mainly rely on users' text inputs to generate potential suggestions. However, images contain rich information that can be quickly perceived. There are some situations where users can imagine what they desire but find it challenging to express it concisely in words. For example, imagine a scenario where a user's bicycle breaks down while riding on the street. In such a case, the intuitive search for the user would be to quickly take a photo of the bicycle to query for a solution rather than relying on TQS or VQS to describe the current issue in text. If the user types "bicycle" in the search box, the suggestions provided may be "bicycle poker", "bicycle shop", and "bicycle pump", which are all irrelevant in expressing the user's intent. In addition, to further enhance the query suggestions, it is desirable for the system to not only provide guidance on fixing a broken bicycle but also offer other useful information, such as nearby bicycle repair stalls and possible reasons why his/her bicycle frequently breaks. These diverse choices allow users to explore the information they may need effectively (as illustrated in Figure 1(c)).

Motivated by practical scenarios, we introduce a novel query formulation, called Multimodal Query Suggestion (MMQS). It takes

a user query image as input and generates query suggestions to response to the user's search intent. Given that the query suggestions are intended to assist users in activating search engines, the design of MMQS focuses on two essential properties:

- Intentionality: The primary goal of MMQS is to capture the user's search intent effectively. Visual data presents an opportunity to infer implicit information needs that might be challenging to articulate in words. By incorporating visual cues from user query images, MMQS aims to provide query suggestions that accurately reflect the user's underlying intent and support more focused and relevant searches.
- Diversity: MMQS generates query suggestions that encompass different aspects of the query image, thereby expanding the search space. This empowers users to explore multiple aspects of information discovery, enhancing the overall search experience.

**Challenges and a New Solution.** The formulation of the MMQS problem introduces several challenges that need innovative solutions. Data Collection (**C1**): Integrating multimodal data comprising both textual and visual information poses unique data preparation challenges. Specifically, it involves generating image-suggestion pairs, a property not presents in many publicly available image-text datasets (e.g., COCO Captions [29] or Flickr30k Entities [34]). Moreover, annotating user intent can be time-consuming and lacks clear guidelines. Therefore, developing efficient and effective strategies for data collection, automated pairing, and reliable annotation becomes crucial for the success of MMQS. Capturing Intentionality and Diversity (**C2**): Inferring user intent from a query image and generating diverse suggestions is a complex task. It requires understanding the visual context and associations between images and textual suggestions. Achieving both intentionality and diversity meanwhile in the generated suggestions necessitates carefully designed techniques to align with user intent and avoid redundancy.

To address the aforementioned challenges, we propose a novel RL4Sugg framework, leveraging the capabilities of Large Language Models (LLMs) with Multi-Agent Reinforcement Learning to generate query suggestions based on input images. To tackle **C1**, we leverage the current GPT language generation capabilities to automate the collection of image-suggestion pairs and user intent annotations based on potential clicks. We employ a threshold-based mechanism that selectively involves manual effort for suggestions with lower confidence scores, ensuring high-quality annotations while striking a balance between automation and human input in the data labeling process. To tackle **C2**, we study a novel solution based on multi-agent reinforcement learning, where we employ two distinct agents within the framework: Agent-I, responsible for intentionality, and Agent-D, responsible for diversity. Specifically, the Agent-I first generates a set of intentional candidate suggestions, which incorporates a RewardNet and a PolicyNet tailored for this task. The RewardNet utilizes multi-task learning to align image-suggestion pairs and assigns rewards to these pairs. Following this, the PolicyNet is trained through Reinforcement Learning from Human Feedback (RLHF) to enhance the intentionality of the suggestions. Further, the Agent-D selects diverse suggestions from the candidate pool, which is designed to cooperate with the Agent-I to ensure that both intentionality and diversity are optimized explicitly in an end-to-end training.

Our contributions can be summarized as follows:

- **The MMQS Task:** We introduce a novel query formulation, called Multimodal Query Suggestion (MMQS), which addresses the gap between multimodal data and query suggestions in search engines. Our objective is to improve the user search experience by providing intentional and diverse query suggestions generated from user query images. To the best of our knowledge, this work presents the first attempt in its kind.
- **The RL4Sugg Framework:** We present a novel framework called RL4Sugg, which is designed to generate query suggestions using user input images. By leveraging the capabilities of LLMs and multi-agent reinforcement learning, RL4Sugg optimizes the intentionality and diversity of the generated suggestions through an end-to-end training.
- **Comprehensive Experiments:** We conduct extensive experiments on two real-world datasets and achieve promising results than various baselines. Our experiments demonstrate the effectiveness of our proposed framework in generating intentional and diverse query suggestions (e.g., it demonstrates 18% improvement compared to the best baseline method). In addition, the proposed MMQS has been transferred into products, and the results show that the deployed system effectively enhances user engagement of search engines.

## 2 RELATED WORK

**Query Suggestion.** Query suggestion is a feature of search engines that provides users with a list of possible queries based on their current query inputs. We review the literature in terms of Textual Query Suggestion (TQS) and Visual Query Suggestion (VQS). For TQS, it relies on the text of the user's query to generate a list of possible textual queries. There are a number of different methods for generating the query suggestions, including (i) query auto completion [2, 39], (ii) query spelling correction [17], (iii) query expansion [5], and (iv) query rewriting [14, 15]. Overall, TQS does not use any visual information, such as images, to generate suggestions.

For VQS, it is introduced by Zha et al. [50, 51], which offers users both textual and visual suggestions based on their query text. This enables users to conveniently specify their search intentions. When a user selects a text-image pair from the suggestion list, the VQS system performs an image search using the provided text and employs the selected image to filter initial search results by leveraging its visual content. Subsequently, many techniques are proposed for the VQS. For example, Zeng et al. [49] develop a new client-side photo search system, which uses VQS and joint text-image hashing to improve the search accuracy and efficiency. Li et al. [28] study video search, and a multimodal method is developed to process the joint text and images suggestions produced by VQS. Overall, our MMQS problem differs from VQS mainly in that the user's query input is different. In MMQS, the input is images, while in VQS, it is text. Additionally, Bian et al. [3] study a new setting of VQS called Visual Query Attributes Suggestion (VQAS), where an image is inputted and VQAS suggests informative attributes (e.g., color, texture, shape) extracted from the query image via some SVM classifiers. These attributes allow users to select and express more precise search intents. Our work differs from VQAS in two aspects. First, MMQS outputs query suggestions instead

of those image attributes, where the suggestions need satisfying the intentionality and diversity properties. Second, we propose a multi-agent reinforcement learning based framework to generate the suggestions from large language models instead of choosing those pre-defined attributes using the classifiers.

**Vision-Language Pre-training.** Our work is related to Vision-Language Pre-training (VLP) in techniques. VLP aims to train a multimodal foundation model to align the relationships between images and text, and then the model is used to support various downstream vision-and-language tasks (e.g., image captioning or visual question answering). The literature on VLP training strategies can be categorized into three main approaches: end-to-end pre-training [7, 18, 21, 25–27, 35, 41, 42], modular pre-training [1, 6, 12, 16, 25, 30, 52, 53], and zero-shot [38, 44, 47].

Our work falls into the modular pre-training, where it makes use of off-the-shelf pre-trained models, keeping them frozen during the pre-training. Existing studies can be categorized according to different frozen components, including the approaches that freeze image encoders [52, 53], language models [6, 12, 16], and both [1, 25, 30]. Specifically, Zhai et al. [52] study Locked-image Tuning (LiT), where it fine-tunes language models via contrastive learning to extract useful representations from locked pre-trained image models for new vision tasks. Driess et al. [12] propose embodied language models, which integrate visual information through a projector into language models. It freezes the language model, and just trains the image encoder with the projector for robotics tasks. Flamingo [1] freezes both image encoders and language models, and introduces cross-attention layers into the language model to incorporate visual features during the fine-tuning. Similarly, BLIP-2 [25] introduces an adapter called Q-Former, which injects visual features into the language model. Our RL4Sugg freezes both image encoders and language models, where we introduce two lightweight agents for fine-tuning, which align the input image to generate query suggestions with RLHF.

**Reinforcement Learning from Human Feedback.** Reinforcement Learning from Human Feedback (RLHF) is an active research area that focuses on training RL agents using human-generated feedback, which is originally developed for training simple robots to interact with real-world environments for complex tasks such as Atari games [9]. Recently, RLHF has been applied to fine-tune various language tasks including text summarization [45], dialogue systems [19, 48], machine translation [22], semantic parsing [24], and review generation [8]. For example, InstructGPT [33] collects a dataset of model desired outputs written by human labelers, and it then adopts RLHF to fine-tune GPT-3 [4]. In this paper, we propose a novel multi-agent reinforcement learning framework, which incorporates RLHF to generate human intentional query suggestions. To our best knowledge, this is the first of its kind.

## 3 PROBLEM STATEMENT

We study the problem of Multimodal Query Suggestion (MMQS), which is formulated below.

PROBLEM 1 (MMQS). *Given a user query image, denoted as $I$, MMQS aims to recommend textual suggestions, denoted as $S =< S_1, S_2, ..., S_K >$. The suggestions are used to help users activate search engines, and thus they need to meet the following two properties:*

- *Intentionality: the suggested queries should align with the content depicted in the query image, and effectively capture the user's intent to offer meaningful options for initiating the search.*
- *Diversity: the suggested queries should reflect different aspects of the query image, offering users a diverse set of choices and avoiding redundancy among them.*

By fulfilling these properties, MMQS aims to enrich the user experience by offering intentional and diverse query suggestions derived from the input query image. MMQS provides a foundational feature for supporting two types of search engines: generation-based and retrieval-based (to be introduced in Section 4.5).

## 4 METHODOLOGY

### 4.1 Overview of RL4Sugg

The proposed solution RL4Sugg addresses the problem of Multimodal Query Suggestion (MMQS) by generating intentional and diverse query suggestions based on user query images. It consists of several key components, including data collection (Section 4.2), Agent-I training (Section 4.3), and Agent-D training (Section 4.4). The overall framework is shown in Figure 2.

In data collection, the language generation capabilities of LLMs are utilized to automate the collection of image-suggestion pairs and the annotation of user intents. This approach combines the efficiency of LLM automation and the reliability of human annotation together to ensure data quality for training. In Agent-I, it generates candidate suggestions by combining a RewardNet and a PolicyNet to capture intentionality. The RewardNet is trained using annotated image-suggestion pairs to assign scores (rewards) indicating the user interest in clicking suggestions. This involves a multi-task learning approach optimizing three pre-training tasks to generate informative rewards. The PolicyNet adopts a two-tower structure to capture visual and textual features and incorporates a Language Model (LLM) to enhance understanding and generation capabilities. It formulates the Markov Decision Process (MDP) for generation, refined through Reinforcement Learning from Human Feedback (RLHF) to ensure alignment with user intents. In Agent-D, it leverages lightweight neural networks to select diverse suggestions from the candidate pool provided by Agent-I, whose MDP is designed so that the two agents cooperatively optimize the both intentionality and diversity of the output suggestions in an end-to-end manner.

We explain some insights behind the RL4Sugg design as follows. (1) RL4Sugg is built based on the combination of LLM automation and human annotation for preparing the training data. It simplifies the data collection process, and reduces the reliance on human annotators for RLHF. (2) The multi-task learning in the RewardNet and RLHF in the PolicyNet enable the Agent-I to learn from various tasks and user feedback, leading to improved performance in generating user intentional suggestions. (3) The Agent-D is trained to minimize the similarity between output suggestions, which ensures that the output suggestions are informative and provide various search aspects for users. Further, Agent-D and Agent-I are trained cooperatively to ensure that the output maintains both intentionality and diversity. This is achieved by optimizing both intentionality and diversity explicitly with multi-agent reinforcement learning.

**Table 1: A running example of data collection. Step 1: GPT-4 generates multiple candidate suggestions from a query image. Step 2: The model assigns a label (1 or 0) to each suggestion, indicating user click intent, along with a confidence score (0 to 1). Step 3: Suggestions with low confidence are filtered out using a confidence threshold (e.g., 0.5) and then undergo human annotation to produce the final labels.**

| | Step 1 | Step 2 | | Step 3 | | |
|---|---|---|---|---|---|---|
| Query Image | Suggestions (generated by GPT) | GPT Labels | Conf | Thres (0.5) | Human Labels | Final Labels |
| | How to fix a broken bicycle chain | 1 | 0.7 | √ | - | 1 |
| | Bicycle chain cleaning | 1 | 0.3 | × | 0 | 0 |
| | Bicycle brand rankings | 0 | 0.6 | √ | - | 0 |
| | Nearby bicycle repair stalls | 1 | 0.8 | √ | - | 1 |
| | Mountain bike prices | 1 | 0.4 | × | 0 | 0 |

## 4.2 Data Collection

This process involves collecting image-suggestion pairs and annotating user intents regarding their likelihood to click on the suggestions or not. However, relying solely on human crowd-sourcing for data collection can be time-consuming and lack clear guidelines. To address this, inspired by language generation capabilities from recent GPT models [13, 30, 32], we propose a novel approach using GPT-4 to automate image-suggestion pair collection and user intent annotation based on potential clicks. This approach provides a balance between automation (by GPT-4) and manual effort (by human annotators) through a threshold-based mechanism. To better illustrate the labeling process, we present a running example in Table 1, which involves three key steps, and the detailed descriptions are included in Appendix Section A.1.

We note that the proposed labeling approach offers several novel aspects in the field of text annotation tasks [13, 30, 32]. First, by utilizing GPT-4's language generation capabilities, we can generate a wide range of candidate suggestions based on image content, providing a comprehensive set of options for users. Second, the labeling and confidence estimation step enhance the reliability of the generated suggestions by quantifying the model's confidence. Third, the threshold-based mechanism introduces a customizable parameter, which facilitates the workload adjustment between automation and human effort according to specific requirements.

## 4.3 Agent-I: Generating Intentional Candidate Suggestions

### 4.3.1 RewardNet.
In this section, we introduce the training process of the RewardNet, utilizing the annotated image-suggestion pair data. The RewardNet provides rewards (e.g., a value ranging between 0 and 1) for each image-suggestion pair, indicating the likelihood of user interest in clicking the suggestion for a given query image. Below, we present the model architecture and training details for the RewardNet.

**Model Architecture.** As shown in Figure 2, our RewardNet employs a Q-Former structure [25], which incorporates an Image-Tower and a Text-Tower, both utilizing transformer-based modules with shared self-attention layers to capture visual and textual features. In the Image-Tower, it incorporates a pre-trained frozen image encoder to extract visual features. To achieve this, we introduce learnable query embeddings as inputs, enabling interactions between queries via self-attention layers and with frozen image features through cross-attention layers. In the Text-Tower, textual suggestions interact with learnable query embeddings through shared self-attention layers.

**Training Paradigm.** We adopt multi-task learning for the RewardNet, optimizing three pre-training tasks: Image-Suggestion Alignment (ISA), Image-Suggestion Generation (ISG), and Image-Suggestion Matching (ISM). The rationale behind the approach is to enhance the RewardNet's training process, facilitating the generation of informative rewards guided by these typical tasks.

In ISA, the goal is to align image and suggestion representations to bring similar pairs closer and push dissimilar ones apart. This is achieved through a contrastive approach. We sample a batch of image-suggestion pairs, each with a label of 1. (2) For each pair $<I_i, S_i>$, we represent them as vectors $\mathbf{v}_i^I$ and $\mathbf{v}_i^S$ via two towers. We treat $\mathbf{v}_i^S$ as the positive of $\mathbf{v}_i^I$ (the anchor), because $I_i$ and $S_i$ have a label of 1, and other suggestions in the batch are considered as the negatives. Then, let $\mathcal{L}_{I,S}$ denote a contrast, which encourages the suggestions to align with the anchor image by comparing their positive and negative pairs, that is,

$$\mathcal{L}_{I,S} = \sum_{<I_i, S_i> \in \mathcal{V}} -\log \frac{\exp\left(\max_{\mathbf{v}_i^I \in \mathbf{V}_i^I} \mathbf{v}_i^I \cdot \mathbf{v}_i^S / \tau\right)}{\sum_{<I_j, S_j> \in \mathcal{V}, j \neq i} \exp\left(\max_{\mathbf{v}_i^I \in \mathbf{V}_i^I} \mathbf{v}_i^I \cdot \mathbf{v}_j^S / \tau\right)}, \quad (1)$$

where $\tau$ represents a temperature parameter. To determine the image-text similarity, we compute the pairwise similarity between each query embedding $\mathbf{v}_i^I \in \mathbf{V}_i^I$ and $\mathbf{v}_i^S$, and select the highest similarity value. Symmetrically, we can define $\mathcal{L}_{S,I}$ by anchoring at $\mathbf{v}_i^S$, then the loss $\mathcal{L}_{ISA}$ is defined as

$$\mathcal{L}_{ISA} = (\mathcal{L}_{I,S} + \mathcal{L}_{S,I})/2. \quad (2)$$

In ISG, the goal is to generate suggestions based on the underlying image content, thereby enhancing the RewardNet's ability to accurately assign scores to image-suggestion pairs. This is achieved by ensuring that the generated suggestions are semantically consistent with the visual context of the grounded image. Specifically, given an image-suggestion $<I, S>$ pair, where the suggestion $S$ corresponds to a sequence of word tokens $S = <\mathbf{w}_1, ..., \mathbf{w}_m>$, we employ a language generation loss to maximize the conditional probability $P$ as

$$\mathcal{L}_{ISG} = \sum_i -\log P(\mathbf{w}_i | \mathbf{w}_{1:i-1}, I). \quad (3)$$

In ISM, the goal is to establish a precise alignment between image and suggestion representations through fine-grained learning. This

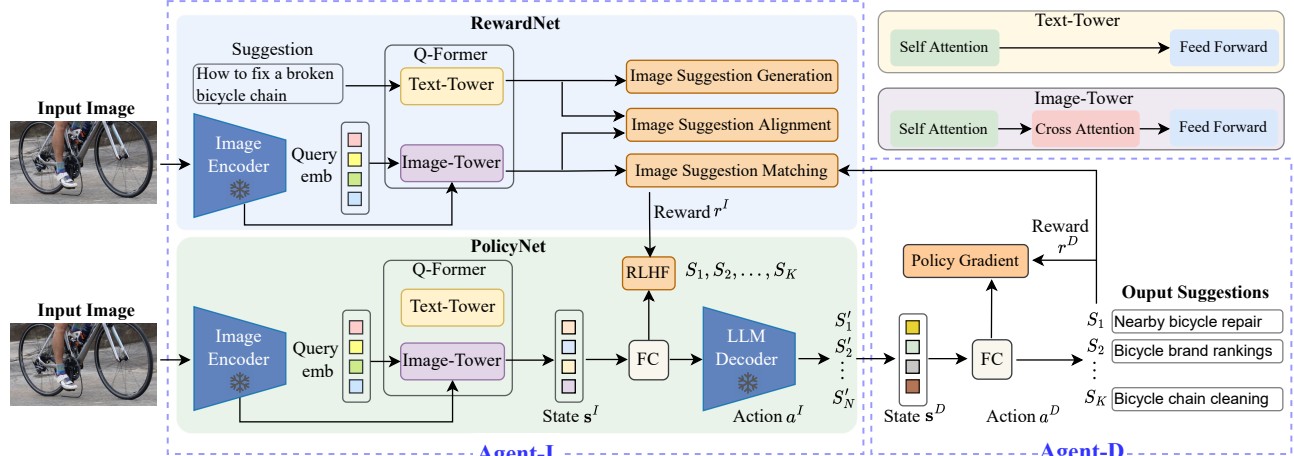

**Figure 2: Training overview of Agent-I and Agent-D. Agent-I trains the RewardNet on three tasks (ISA, ISG, ISM) using learnable query embeddings, while the PolicyNet is trained with RLHF to generate candidate suggestions $S_1', S_2', ..., S_N'$ for intentionality. Agent-D learns to select diverse suggestions from the candidates via policy gradient and outputs the final $K$ suggestions.**

involves a binary classification task in which the model is to predict whether an image-suggestion pair is positive (matched) or negative (unmatched). To achieve this, we use a hard negative mining strategy, where hard negative samples are image-related suggestions labeled as 0. The rationale is that while some suggestions are related to the query image, they fail to capture the user's search intent. By optimizing with these hard samples, the RewardNet is encouraged to assign high scores to the pairs exhibiting a strong intention. Then, the objective is trained using a binary cross-entropy loss, formulated as

$$\mathcal{L}_{\text{ISM}} = -y * \log(P) + (y - 1) * \log(1 - P), \quad (4)$$

where $y$ denotes the true label (either 0 or 1), and $P$ is the predicted probability of the positive class.

Finally, the RewardNet is trained using a multi-task learning approach, where the loss function $\mathcal{L}_{\text{RN}}$ is defined as

$$\mathcal{L}_{\text{RN}} = \mathcal{L}_{\text{ISA}} + \mathcal{L}_{\text{ISG}} + \mathcal{L}_{\text{ISM}}. \quad (5)$$

Note that the reward is then obtained as the predicted probability $P$ in the ISM task, where it scores a normalized value ranging from 0 to 1, which avoids potential data scale issues that may arise during the training process, that is

$$r_\theta(I, S) = P, \quad (6)$$

where $r_\theta(I, S)$ denotes the reward for a given query image $I$ and its associated suggestion $S$, and $\theta$ denotes the RewardNet parameters.

4.3.2 **PolicyNet.** The objective of MMQS is to generate query suggestions that align with users' intended search queries, specifically those that are more likely to be clicked. This motivates us to explore the application of Reinforcement Learning from Human Feedback (RLHF) technique in training the PolicyNet. Below, we present the model architecture, and MDPs in the PolicyNet.

**Model Architecture.** In PolicyNet, we adopt a similar two-tower structure as presented in the RewardNet, to capture both visual and textual features. Additionally, we aim to leverage the language

generation capability of a LLM by establishing a connection between the Image-Tower and the LLM. As shown in Figure 2, the connection is implemented using a fully-connected (FC) layer. The FC layer projects the output query embeddings to align with the dimensionality of the LLM's text embedding, and then these projected query embeddings are concatenated at the beginning of the input text embeddings of the LLM. This integration serves the visual information as soft prompts, conditioning the LLM on the visual representations to generate language. Notably, the LLM is kept frozen during training to facilitate the process.

**MDP for Generating Suggestions.** To enhance the intentionality of the generated suggestions, we model the process with RLHF, involving states, actions, and rewards.

*States*: The state $\mathbf{s}^I$ is defined by the learned query embeddings of an input query image, which undergoes a process to extract the representation. Specifically, the image is first encoded using a frozen Vision Transformer (ViT) [35], which produces a fixed-length representation of the image that captures its visual features. Then, some learnable query embeddings are generated as the design in RewardNet, these embeddings represent the different aspects of the query image that the model should attend to, and the query embeddings are then passed through cross-attention layers, which allow them to interact with the frozen visual features. By leveraging this approach, we can effectively incorporate the contextual relationships between the queries and the image features, and forming a comprehensive representation of the state.

*Actions*: The action $a^I$ is defined by the generated suggestions via a LLM, which conditions on the state representation to generate language. Here, We employ a decoder-only language model (e.g., OPT [54]) for its simplicity and efficiency, as it does not require encoding input information, and only to generate suggestions that are relevant to the image. This enables our training more efficiently and reduces GPU requirements.

*Rewards*: The reward $r^I$ is obtained from the RewardNet according to Equation 6. The purpose of training the reward model is to accurately predict the quality of a generated suggestion, as assessed by human judgment. It is important to note that Agent-I's

action involves exploring candidate suggestions, and the reward cannot be immediately observed because the final suggestions have not yet been generated. When the action is to provide the candidates for Agent-D to choose final suggestions within this candidate pool, some reward signal can be acquired (e.g., measuring the intentionality of suggestions). Subsequently, the PolicyNet would be updated accordingly through RLHF (more training details are presented in Section 4.4). This approach facilitates the cooperation between Agent-I and Agent-D, guiding them towards the joint goal of producing intentional and diverse suggestions in the final output.

## 4.4 Agent-D: Choosing Diverse Suggestions from the Candidates

**MDP for Choosing Suggestions.** We further introduce an Agent-D to enhance the overall diversity of suggestions and provide users with a more comprehensive selection. We discuss the rationale behind the introduction of this agent. One straightforward method to increase diversity is to employ post-processing techniques like clustering. This technique groups similar candidate suggestions into clusters and selects the cluster centers as output to reduce redundancy. However, such post-processing faces two challenges: (1) the model cannot directly generate both intentional and diverse suggestions, which makes further optimization difficult; (2) the clustered suggestions prioritize diversity but may sacrifice intentionality in the output. To tackle the challenges, we consider the diversity as one of the training objectives managed by Agent-D, where it calculates semantic similarity between suggestions, and cooperative training with Agent-I during the policy training process. This end-to-end optimization empowers the language model to generate suggestions that exhibit both intentionality and diversity.

To accomplish this task, we use a sliding window algorithm with a window size denoted as $K$. The candidate suggestions provided by Agent-I are represented as $< S'_1, S'_2, ..., S'_N >$, and Agent-D's objective is to select the $K$ diverse suggestions from this set (where $K < N$). Here is how the sliding window algorithm operates: (1) The algorithm begins by scanning the first $K$ suggestions and deciding which one within the window should be omitted. (2) It then inserts the next suggestion into the window and repeats the decision-making process. (3) This scanning and decision-making continue until all suggestions have been processed. (4) Finally, the algorithm maintains and outputs the best $K$ suggestions during the scanning, which correspond to the highest diversity. Diversity is measured by computing pairwise semantic similarities among the $K$ suggestions $< S_1, S_2, ..., S_K >$, typically involving a subtraction operation (where a larger diversity implies smaller similarity), i.e.,

$$DIV = \frac{1}{2} - \frac{\sum_{1 \le i < j \le K} \sigma(S_i, S_j)}{K * (K - 1)}, \tag{7}$$

where $\sigma(\cdot, \cdot)$ represents a similarity measurement between two suggestions, typically calculated using methods like cosine similarity with S-BERT [36]. This similarity score is then normalized to a value between 0 and 1 for clarity. Below, we introduce the MDP of Agent-D, which decides the process of selecting which suggestions to drop from the window. This decision-making process is guided by lightweight fully-connected (FC) neural networks trained through the policy gradient method [40, 43].

*States:* In the context where we have $N$ candidate suggestions denoted as $< S'_1, S'_2, ..., S'_N >$, we utilize S-BERT embeddings [36] to capture their semantic features, which are represented as $\mathbf{b}^S_i$ for each suggestion ($1 \le i \le N$). The state $\mathbf{s}^D$ is defined by concatenating these $N$ embeddings, i.e., $\mathbf{s}^D = \{\mathbf{b}^S_1, \mathbf{b}^S_2, ..., \mathbf{b}^S_N\}$.

*Actions:* We denote an action of Agent-D as $a^D$, and the design of these actions is based on the previous discussion, which involves dropping one of the $K$ suggestions in the sliding window and inserting the next suggestion into the window. Formally, the actions are defined as $a^D = k$ where $1 \le k \le K$. In this notation, when action $a^D = k$, it means that the $k$-th suggestion should be dropped, and the $K + 1$-th suggestion should be inserted into the window. Consider the consequence of dropping the $k$-th suggestion, this action transitions the environment to the next state as $\mathbf{s}'^D = \{\mathbf{b}^S_1, ..., \mathbf{b}^S_{k-1}, \mathbf{b}^S_{k+1}, ..., \mathbf{b}^S_K, \mathbf{b}^S_{K+1}, ..., \mathbf{b}^S_N, \mathbf{O}\}$, where $\mathbf{O}$ represents a zero vector, which is used to pad the state $\mathbf{s}'^D$ into a fixed-length vector. This fixed-length vector is then fed into the fully-connected (FC) policy network.

*Rewards:* We denote the reward as $r^D$. The reward associated with the transition from state $\mathbf{s}^D$ to state $\mathbf{s}'^D$ after taking action $a^D$ is defined as: $r^D = \mathbf{s}'^D.DIV_{best} - \mathbf{s}^D.DIV_{best}$, where $\mathbf{s}^D.DIV_{best}$ represents the maintained best diversity value found at state $\mathbf{s}^D$ during the scanning according to Equation 7. With this reward definition, the objective of the MDP, which is to maximize the cumulative rewards, aligns with the goal of discovering the greatest diversity among the suggestions. To illustrate this alignment, consider the process as it moves through a sequence of states: $\mathbf{s}^D_1, \mathbf{s}^D_2, ..., \mathbf{s}^D_N$, ending at $\mathbf{s}^D_N$. We can denote the rewards received at these states, except for the termination state $\mathbf{s}^D_N$, as $r^D_1, r^D_2, ..., r^D_{N-1}$. When future rewards are not discounted, we have:

$$\sum_{t=2}^{N} r^D_{t-1} = \sum_{t=2}^{N} (\mathbf{s}^D_t.DIV_{best} - \mathbf{s}^D_{t-1}.DIV_{best}) \tag{8}$$
$$= \mathbf{s}^D_N.DIV_{best} - \mathbf{s}^D_1.DIV_{best},$$

where $\mathbf{s}^D_N.DIV_{best}$ corresponds to the highest diversity value found during the scanning process, and $\mathbf{s}^D_1.DIV_{best}$ represents the initial diversity value, which remains constant. Consequently, maximizing the cumulative rewards is equivalent to maximizing the diversity that can be discovered.

**Learning Policies of Agent-I and Agent-D.** We discuss the learning process of the two agents.

For Agent-I, to train the PolicyNet, which involves two stages: (1) warm-start stage and (2) training stage. In (1), we study the Supervised Fine-Tuning (SFT), which equips the LLM with the basic abilities to generate suggestions, where the two towers of the PolicyNet are trained using a multi-task learning approach (ISA, ISG, and ISM) according to Equation 4, which allows them to learn from different related tasks simultaneously. In (2), we utilize the PPO algorithm [37] to fine-tune the SFT model for achieving the intentionality, where the environment is modeled as a bandit setting, i.e., when a random query image is presented, the model generates a suggestion in response to the image, and ends the episode. The loss contains the following components: i) Following the output suggestions (denoted by $< S_1, S_2, ..., S_K >$) by Agent-D, the environment calculates a reward $r^I$ via the RewardNet according

to Equation 6. ii) In addition, we fine-tune the connection (i.e., the FC layer) between the LLM and the two-tower using a language generation loss on the output suggestions according to Equation 3. By conditioning the LLM on the output from the two-tower to generate language, it can capture the visual cues presented in the input image. iii) To prevent over-optimization of the RewardNet, we further incorporate a penalty for the KL divergence [19] between the learned RL policy, denoted as $\pi_\phi^{\text{RL}}$ with parameters $\phi$, and its original SFT policy, denoted as $\pi^{\text{SFT}}$. Formally, the loss of Agent-I is presented as

$$\mathcal{L}_{\text{I}} = -r^I + \beta \log(\pi_\phi^{\text{RL}}(a^I|\mathbf{s}^I)/\pi^{\text{SFT}}(a^I|\mathbf{s}^I)) - \gamma \sum_i \log P(\mathbf{w}_i|\mathbf{w}_{1:i-1}, I),$$ (9)

where $\beta$ and $\gamma$ are two coefficients to control the strength of the KL penalty and language loss. For each output suggestion $S_i$ ($1 \leq i \leq K$), it corresponds to a sequence of word tokens $S_i = < \mathbf{w}_1, ..., \mathbf{w}_m >$ for the language generation.

For Agent-D, the core problem of its MDP is to acquire a policy that guides the agent in selecting actions denoted as $a^D$. These actions are determined based on the constructed states $\mathbf{s}^D$ with the objective of maximizing the cumulative reward, denoted as $R_N$. We employ a policy gradient method for learning this policy, called the REINFORCE algorithm [40, 43]. To elaborate, we introduce a stochastic policy denoted as $\pi_\theta(a^D|\mathbf{s}^D)$. This policy is responsible for probabilistically sampling an action $a^D$ for a given state $\mathbf{s}^D$ using a neural network, where the network's parameters are represented as $\theta$. The loss function for Agent-D is then formulated as follows:

$$\mathcal{L}_{\text{D}} = -R_N \ln \pi_\theta(a^D|\mathbf{s}^D).$$ (10)

## 4.5 Discussion on Applications and Cold-start

**Supporting Generation-based and Retrieval-based Applications.** We explore RL4Sugg capabilities in two search engine scenarios: (1) generation-based and (2) retrieval-based. In (1), RL4Sugg can naturally generate query suggestions using its language generation capability from LLMs in response to users' image queries across diverse domains. In (2), RL4Sugg specializes in providing query suggestions for specific domains with narrower focuses, like E-commerce shopping websites, where the query suggestions are limited to their commodities, and can be prepared in advance. It leverages its ability to represent images and language in PolicyNet's two-tower. Query suggestions are stored as vector representations in a database, and vector-based retrieval, such as HNSW, enhances search efficiency. During inference, RL4Sugg extracts the user's image representation and retrieves Top-$K$ query suggestions with high similarity. Notably, this approach offers several advantages, including efficient query response, and by precomputing and storing the query suggestions in a database, the quality of these suggestions can be guaranteed in advance.

**Handling the Cold-start Problem.** Since RL4Sugg relies on annotator feedback to understand search intentionality, RL4Sugg faces a potential cold-start issue for recommending suggestions when the learned knowledge is insufficient for online user queries. To tackle this issue, we employ online learning to continuously fine-tunes both agents by Equation 9 and 10, using newly recorded query images and user-clicked suggestions (i.e., labeling as 1), ensuring the model's policy remains up-to-date for online use. In Section 5.2, we validate this approach, and the results show significant improvements by 8.3% in user experience, which indicates the positive impact of this strategy in practice.

## 5 EXPERIMENTS

### 5.1 Experimental Setup

**Dataset and Ground Truth.** We conduct experiments on two real-world datasets: Business and ImageNet [11]. The Business dataset contains around 50,000 user query images collected from a real image search engine between January 2022 and January 2023. We randomly sample 80% of these images for training, and the remaining for testing. For each image, we collect 5 suggestions following the data collection process described in Section 4.2, where 46.9% suggestions are labeled by the GPT model, and the remaining suggestions are labeled by 20 human labelers. Among them, 75.8% image-suggestion pairs are with the label 1. Similarly, we collect 1,000 image-suggestion pairs with labels from the ImageNet, which are used to test the transferability of the model fine-tuned on the Business and to perform zero-shot evaluations on the ImageNet.

By following [33], we then discuss the ground truth for evaluation, considering both the retrieval and generation tasks. For the retrieval task, we establish the ground truth of each query image by considering its suggestions with a label of 1. For quality control, we randomly pick 10% labeled image-suggestion pairs, ask 5 other checkers to label these suggestions independently. We employ majority voting to aggregate the labels, and compute the accuracy (denoted by $\delta$) of the labels by the labelers against the aggregated ones by the checkers. The $\delta$ is 76.7% for the Business and 78.3% for the ImageNet, which demonstrate the high accuracy of our evaluations. For the generation task, we let the human labelers to assess the suggestions generated by various baseline methods and RL4Sugg. These labeled suggestions are then reviewed by 5 other checkers. Similarly, we report the $\delta$ values as a measure of quality verification.

**Baseline.** We carefully examine existing vision-language models, and identify the following baseline methods: Flamingo, BLIP-2, LLaVA for the generation task, and CLIP, BLIP-2 for the retrieval task. These methods have comparable parameter sizes of LLM backbones as our OPT$_{2.7\text{B}}$ for addressing the MMQS problem. Notably, these models are open-sourced, and we fine-tune them using our collected image-suggestion pairs for fair comparisons. The details are introduced in Appendix Section A.2 due to the page limit.

**Implementation Details.** The implementation details of RL4Sugg and training process can be found in Appendix Section A.3 due to the page limit.

**Evaluation Metrics.** We evaluate the RL4Sugg in terms of the generation task and the retrieval task. For the generation task, We report Discounted Cumulative Gain (DCG) and Good vs. Same vs. Bad (GSB) by following [10, 31]. For the retrieval task, we report positive-negative ratio (PNR) and Recall@K by following [20, 31]. In addition, We report the DIV according to Equation 7 for measuring the diversity within a set of output query suggestions. Overall, superior results are indicated by higher values of DCG, GSB, PNR,

**Table 2: Effectiveness of generation-based applications, fine-tuned on Business and zero-shot transferred to ImageNet, where $\delta$ indicates the accuracy of labeling the generated suggestions as introduced in Section 5.1.**

| Models | #Train/#Total Params | Business Fine-tuned | | | ImageNet 0-Shot | | |
|---|---|---|---|---|---|---|---|
| | | DCG | DIV | $\delta$ | DCG | DIV | $\delta$ |
| Flamingo | 1.4B/3.4B | 0.73 | 0.25 | 81.7% | 0.67 | 0.23 | 80.6% |
| BLIP-2 | 104M/3.1B | 0.59 | 0.17 | 68.3% | 0.47 | 0.18 | 69.2% |
| LLaVA | 14M/13B | 0.60 | 0.25 | 73.3% | 0.47 | 0.24 | 76.5% |
| RL4Sugg | 208M/3.1B | **0.89** | **0.25** | **83.3%** | **0.87** | **0.24** | **86.9%** |

**Table 3: Effectiveness of retrieval-based applications, fine-tuned on Business and zero-shot transferred to ImageNet.**

| Models | #Train/#Total Params | Business Fine-tuned | | | ImageNet 0-shot | | |
|---|---|---|---|---|---|---|---|
| | | PNR | R@1 | R@3 | PNR | R@1 | R@3 |
| CLIP | 300M/300M | 1.30 | 0.23 | 0.33 | 0.90 | 0.21 | 0.32 |
| BLIP-2 | 104M/3.1B | 1.05 | 0.27 | 0.60 | 0.73 | 0.26 | 0.58 |
| RL4Sugg | 208M/3.1B | **2.80** | **0.63** | **0.83** | **2.17** | **0.58** | **0.74** |

**Table 4: Ablation study (Business).**

| Components | DCG | DIV |
|---|---|---|
| RL4Sugg | **0.89** | **0.25** |
| w/o RLHF (SFT) | 0.78 | 0.24 |
| w/o Agent-D (Agent-I only) | 0.89 | 0.19 |
| w/o Agent-D (greedy) | 0.82 | 0.23 |

Recall@$K$, and DIV. The detailed description is included in Appendix Section A.4 due to the page limit.

## 5.2 Experimental Results

**(1) Effectiveness evaluation (comparison with baseline methods).** We conduct experiments to verify the effectiveness of both generation task and retrieval task, where we fine-tune the model on Business (the number of trainable parameters is reported) and directly test its performance on ImageNet for transferability study. For the generation task, as shown in Table 2, we query 300 images on both Business and ImageNet datasets, where the models generate three suggestions on each image for human evaluation to calculate the DCG, and $\delta$ is also reported to indicate the evaluation accuracy. We observe that the DCG of RL4Sugg outperforms all other baseline models and shows good transferability. The best baseline model is Flamingo with the DCG of 0.73, which is 18% lower than RL4Sugg. In addition, we observe that all models have similar diversity except BLIP-2, because BLIP-2 sometimes generates query suggestions with same meaning expressed by different words, and LLaVA tends to generate longer query suggestions so its diversity is relatively high. Since the query suggestions are all based on query images, which contain some necessary described entities or common grammar structures, the diversity values of all models are not very high in general. For the retrieval task, as shown in Table 3, RL4Sugg shows better PNR and Recall compared with the other two baseline models on both datasets.

**(2) Ablation study.** To evaluate the effectiveness of the two agents in RL4Sugg, we conduct an ablation study. We replace the RLHF in Agent-I and use the SFT model only; we remove the Agent-D, or replace it with a pre-defined rule of dropping the most similar

**Table 5: Online A/B Test (Business).**

| Metric | Cold-start | | |
|---|---|---|---|
| | A (old RL4Sugg) | B (new RL4Sugg) | Impr |
| # Click behaviors | **0.46%** (vs. old RL4Sugg) | | |
| DCG | 0.83 | **0.89** | 6.7% |
| GSB | **8.3%** (vs. old RL4Sugg) | | |
| PNR | 2.61 | **2.80** | 6.8% |
| R@1 | 0.57 | **0.63** | 9.5% |
| R@3 | 0.75 | **0.83** | 9.6% |

suggestion within the window. We present the results in Table 4, which demonstrate that both agents contribute to improving the performance. Specifically, we observe that the DCG drops dramatically without the RLHF training from 0.89 to 0.78, which indicates that RLHF can capture human intentionality. As expected, when we remove the Agent-D, the diversity decreases significantly from 0.25 to 0.19. If we use a rule to greedily drop the suggestions, the diversity also decreases, and we note that the DCG also decreases from 0.89 to 0.82. This is because the rule simply drops those similar query suggestions without considering the intentionality. By incorporating the Agent-D, which interacts with the Agent-I during the training so it guides the Agent-I to generate more diversified query suggestions while preserving the intentionality.

**(3) Parameter study (varying confidence threshold in data collection).** We investigate the effect of varying confidence threshold in data collection on the generation task and the retrieval task. The results and detailed analysis are included in Appendix Section A.6 due to the page limit. Overall, we observe that a moderate threshold can produce good results and save human efforts.

**(4) Online A/B Test.** We conduct an online A/B test to compare the new system (after online learning for the cold-start problem) with the old system for one month. The results as shown in Table 5 demonstrate that the fine-tuned model via online learning can largely improve the overall user experience, e.g., it increases the number of click behaviors by 0.46%. In addition, we collect online cases, and compare the two systems with the real user-generated queries via manual evaluation. We observe that the new system can largely outperform the base system.

**(5) Qualitative results.** We qualitatively evaluate the generated query suggestions. The detailed visualization results and analysis are in Appendix Section A.7 due to the page limit. Overall, we observe that the suggestions align well with user search intents.

## 6 CONCLUSION

In this paper, we introduce a novel Multimodal Query Suggestion (MMQS) framework that addresses the limitations of existing query suggestion systems by incorporating user query images. Through the MMQS approach, we significantly enhance the intentionality and diversity of query suggestions, resulting in a more user-centric and effective search experience. Extensive experiments conducted on two real-world datasets demonstrate a remarkable 18% improvement over the best existing approach. Moreover, our successful deployment of MMQS into real-world products showcases its practicality and potential for providing valuable insights in search engines. As a future direction, we plan to extend MMQS to accommodate other modalities, such as audio or video, to enhance its applicability in diverse real-world scenarios.

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

# A  APPENDIX

## A.1  Data Collection Details

The data collection process involves three key steps, which are presented below:

*Step 1: Candidate Suggestion Generation.* Leveraging extracted information (e.g., captions) from an image, GPT-4 initially employs its language generation capabilities to generate multiple candidate suggestions. These suggestions are designed to capture various aspects of the image, providing a wide range of options that may align with their interests.

*Step 2: Labeling and Confidence Estimation.* Once the candidate suggestions are generated, GPT-4 further proceeds to label each suggestion based on its relevance to the image and potential user intent. The process enables GPT-4 to assign a binary label (i.e., 1 or 0) to each suggestion, indicating the user interest in clicking the suggestion for a given query image or not. In addition, GPT-4 provides a confidence score (ranging from 0 to 1) associated with each label, which serves as an indicator of the model's certainty in its own labels.

*Step 3: Threshold-based Annotation.* To mitigate the reliance on manual annotation while maintaining annotation quality, we introduce a threshold-based mechanism. The confidence scores generated by GPT-4 are used to indicate the quality of the suggested annotations. By setting a threshold value, suggestions with low confidence scores are identified for further manual annotation. This approach reduces the burden of extensive manual annotation while ensuring that suggestions with lower confidence are subject to human review.

## A.2  Baseline Details

We compare RL4Sugg with the following baseline methods, and the details are presented below:

• Flamingo [1]: it freezes the image encoders and language models during the fine-tuning process, and the language model learns to use visual features by adding cross-attention layers.

• BLIP-2 [25]: it bridges the modality gap between image and language with a lightweight Transformer adapter, which trains following a two-stage strategy, i.e., vision-and-language representation learning, then vision-to-language generative learning.

• LLaVA [30]: it studies an automatic strategy for generating language-image instruction-following dataset, and a multimodal model connecting the image encoder and the language model is trained end-to-end based on the dataset.

• CLIP [35]: it leverages contrastive language-image pre-training to produce highly effective image and text representations, which can transfer well to different tasks.

## A.3  Implementation Details

**Agent-I Training.** Our RL4sugg model is composed of two agents: Agent-I and Agent-D. Agent-I has two modules: RewardNet and PolicyNet. Before training these modules, we fine-tune the BLIP-2 model for its two stages to obtain an SFT model on the COCO dataset. Since the COCO dataset has image captions instead of query suggestions, we input these captions to the GPT model as context prompts and ask it to output query suggestions. We then pair up the

**Table 6: Hyperparameters for training SFT.**

| Stages | Stage 1 | Stage 2 |
|---|---|---|
| Pretrained model | blip2-pretrain | blip2-sft_stage1 |
| Epochs | 10 | |
| Learning rate schedule | linear_warmup_cosine_lr | |
| Warmup learning rate | 1e-6 | |
| Initial learning rate | 1e-4 | |
| Minimum learning rate | 1e-5 | |
| Warmup steps | 5000 | 2000 |
| Weight decay | 0.05 | |
| Batch size | 64 | 16 |
| Image resolution | 224 | |

**Table 7: Hyperparameters for training RewardNet.**

| Stages | Stage 1 |
|---|---|
| Pretrained model | blip2-sft_stage1 |
| Epochs | 10 |
| Learning rate schedule | linear_warmup_cosine_lr |
| Warmup learning rate | 1e-6 |
| Initial learning rate | 1e-4 |
| Minimum learning rate | 1e-5 |
| Warmup steps | 5000 |
| Weight decay | 0.05 |
| Batch size | 64 |
| Image resolution | 224 |

**Table 8: Hyperparameters for training PolicyNet.**

| Stages | Stage 2 |
|---|---|
| Pretrained model | blip2-sft_stage2 |
| Epochs | 10 |
| Learning rate schedule | linear_warmup_cosine_lr |
| Warmup learning rate | 1e-6 |
| Initial learning rate | 1e-4 |
| Minimum learning rate | 1e-5 |
| Warmup steps | 2000 |
| Weight decay | 0.05 |
| Batch size | 50 |
| Image resolution | 224 |

**Table 9: Hyperparameters for training Agent-D.**

| Approach | Policy gradient |
|---|---|
| Optimizer | Adam |
| Learning rate | 1e-3 |
| Discount factor | 0.99 |

image and query suggestions and use these image-suggestion pairs to fine-tune the BLIP-2 model. During the two stages of training, we use the checkpoint of stage 1 (resp. stage 2) to initialize the RewardNet (resp. PolicyNet). After initialization, we train RewardNet (resp. PolicyNet) on the Business (resp. Flickr30k) dataset. The Business dataset is constructed in the form of image-suggestion pairs, so it can be used for this training directly. Additionally, we only use images in Flickr30k for this training. Specifically, these images are fed into PolicyNet, which generates 20 query suggestions. These

image-suggestion pairs are then sent to RewardNet to get rewards, which are further used for training PolicyNet.

**Agent-D Training.** Agent-D is a network that consists of three fully-connected layers. The number of neurons in each layer is 512, 128, and 3, respectively. The dropout rate for hidden layers is 0.5, and the activation function is ReLU. It is pre-trained before being integrated into the RL4Sugg model. Specifically, Agent-D first learns to select some diversified suggestions from many candidate suggestions during pre-training. Then, it trains with Agent-I to optimize both intentionality and diversity. During pre-training, we sample 100 sets of candidate suggestions from the training dataset. For each set, we generate 200 episodes for policy gradient. Each episode involves around 20 steps, resulting in approximately 4 million transition steps during the learning process. At each step, we sample an action using the probability outputted by the softmax function at the current state.

**Training time.** Using a machine with 8 Nvidia RTX 3090 (24GB memory), it takes 6 hours to fine-tune the BLIP-2 model to obtain the SFT model and 2 hours for pre-training Agent-D. In Agent-I, RewardNet takes 1.5 hours while PolicyNet requires 4.5 hours for training.

We summarize the hyperparameter settings for training the SFT model, RewardNet, PolicyNet and Agent-D in Table 6, Table 7, Table 8 and Table 9, respectively.

## A.4 Evaluation Metrics

For the generation task, we report Discounted Cumulative Gain (DCG) and Good vs. Same vs. Bad (GSB) by following [10, 31]. For DCG, it is to evaluate the effectiveness of a list of suggestions produced by a system. The DCG is defined as follows:

$$DCG = \sum_{i=1}^{K} \frac{rel_i}{\log_2(i+1)}, \quad (11)$$

where $rel_i$ represents the intentionality of the suggestion (e.g., whether it is clicked or not) at position $i$ ($rel_i \in \{0, 1\}$), and $K$ denotes the returned $K$ query suggestions. For GSB, it involves human experts to determine whether the new system or the base system provides superior final results, where the relative gain is measured using the Good vs. Same vs. Bad (GSB), that is

$$GSB = \frac{\#Good - \#Bad}{\#Good + \#Same + \#Bad}, \quad (12)$$

where $\#Good$ (resp. $\#Bad$) is a counter that increments by 1 if the new system delivers better (resp. worse) results compared to the base system, and $\#Same$ increments by 1 otherwise.

For the retrieval task, we report positive-negative ratio (PNR) and Recall@K by following [20, 31]. For PNR, it is defined as the ratio of positive instances over negative instances for a given query image $I$ and its suggestions $\mathcal{S}$, that is

$$PNR = \frac{\sum_{S_i, S_j \in \mathcal{S}} \mathbb{1}(y_i > y_j) \cdot \mathbb{1}(f(I, S_i) > f(I, S_j))}{\sum_{S_i', S_j' \in \mathcal{S}} \mathbb{1}(y_i' > y_j') \cdot \mathbb{1}(f(I, S_i') < f(I, S_j'))}, \quad (13)$$

where $y_i$ denotes the manual label (i.e., click or not by users) of the suggestion $S_i$, and $f(I, S_i)$ denotes the cosine similarity based on the representations of query image $I$ and suggestion $S_i$. The indicator function $\mathbb{1}(\cdot)$ is used to represent whether a certain condition is true or false, e.g., $\mathbb{1}(y_i > y_j) = 1$ if $y_i > y_j$, and 0 otherwise. Intuitively,

**Table 10: Parameter study (Business), 0 and 1 indicate all suggestions are labeled by GPT and human, respectively.**

| Threshold | 0 | 0.2 | 0.4 | 0.6 | 0.8 | 1 |
|---|---|---|---|---|---|---|
| DCG | 0.83 | 0.85 | 0.86 | 0.89 | 0.89 | 0.90 |
| PNR | 2.40 | 2.60 | 2.70 | 2.80 | 2.83 | 2.85 |
| Recall@1 | 0.54 | 0.58 | 0.61 | 0.63 | 0.64 | 0.64 |
| #Suggs for human labeling | 0 | 58K | 96K | 133K | 202K | 250K |

PNR quantifies the agreement between the manual labels and the model scores. For Recall@$K$, it is defined as

$$Recall@K = \frac{|\mathcal{S}| \cap |\hat{\mathcal{S}}|}{K}, \quad (14)$$

where $\hat{\mathcal{S}}$ denotes the suggestions for a query image $I$ by a retrieval model, and $\mathcal{S}$ denotes the set of ground truth suggestions (e.g., the suggestions that will be clicked by human labelers) for the $I$. We report the average PNR and Recall@$K$ values across all queries in our experiments.

In addition, to measure the diversity within a set of output query suggestions, we report the DIV according to Equation 7. Overall, superior results are indicated by higher values of DCG, GSB, PNR, Recall@$K$, and DIV.

## A.5 Evaluation Platform

We implement RL4Sugg and other baselines in Python 3.7 and PyTorch 1.8.0. The experiments are conducted on a server with 32 cores of Intel(R) Xeon(R) Gold 6151 CPU @ 3.00GHz 512.0GB RAM and 8 Nvidia RTX3090 GPU (24GB memory).

## A.6 Parameter Study of Confidence Threshold

Recall that during data collection, GPT model is used to reduce the workload of human labelers, where the suggestions with lower confidence than a threshold are subject to human labeling. We vary the threshold from 0 to 1, where 0 (resp. 1) indicates all suggestions are labeled by GPT (resp. human). Within the setting, we train 6 versions of RL4SUGG models, and the effectiveness is reported in Table 10. We choose the threshold of 0.6 as the default setting, since the effectiveness is near to the optimal, and it reduces a large amount of labeling effort for 46.9%.

## A.7 Qualitative results

Table 11 demonstrates examples to show a wide range of zero-shot capabilities on image-to-suggestion generation. We choose Flamingo for comparison, since it shows the best effectiveness among baselines. We observe that our query suggestions cover various intentions of the query image. For Case-1, a potential intention could involve the task of cleaning or organizing a dirty fridge. Notably, we observe that RL4Sugg effectively captures this intuitive intention, which demonstrates a commendable level of intentionality following RLHF training. For Case-2, RL4Sugg notices that the user might be interested in the dress for a photoshoot or the outdoor environment, and thus it recommends two suggestions accordingly instead of simply describing the content of the image. However, Flamingo wrongly recognizes "wedding dresses" in the image. For

**Table 11: Examples of zero-shot image-to-suggestion generation using RL4Sugg and Flamingo.**

| No. | Query Image | RL4Sugg | Flamingo | GSB |
|-----|-------------|---------|----------|-----|
| 1 | | Tips for keeping a refrigerator clean
How to organize and clean a fridge | Desirable refrigerator brands
Where to buy shampoo | 1.00 |
| 2 | | How to choose the right dress for a outdoor photoshoot
Benefits of spending time in nature | Desirable wedding dresses
Where to buy wedding dresses | 1.00 |
| 3 | | How to fix a broken iPhone screen
How to clean a broken iPhone screen | Where to buy a new iPhone
Where to buy a new iPad | 0.50 |
| 4 | | How to make fresh breads at home
Best places to buy fresh baked bread in the area | Where to buy breads
Where to buy breads | 0.50 |
| 5 | | Snowboarding safety tips and tricks
How to choose the right snowboarding gear | Desirable snowboard brands
Where to buy snowboard boots | 0.00 |

Case-3, RL4Sugg can accurately captures a high-intention query ("a broken iPhone") similar to Case-1. For Case-4, RL4Sugg provides suggestions with good diversity with the aid of our Agent-D, e.g., it generates suggestions about how to make them or where to buy them; however, Flamingo generates duplicated suggestions ("Where to buy breads") in this case. For Case-5, we observe that Flamingo frequently uses a fixed pattern to process the image query, such as "Desirable something" or "Where to buy something" (as seen in Case-1, Case-2 and Case-5), where it succeeds in recognizing the correct object ("snowboard boots") in the image. However, when users frequently notice the fixed pattern, they might become bored with the recommended suggestions.

