# OpenReview forum: "Multimodal Query Suggestion with Multi-Agent Reinforcement Learning from Human Feedback"
_ACM.org/TheWebConf/2024/Conference — TheWebConf24_

### Official Review · Reviewer_6GYT · 2023-11-27

**Novelty:** 3
**Technical Quality:** 4

**Review:**

In the paper, the authors introduce a novel Multimodal Query Suggestion (MMQS) task, which aims to generate query suggestions based not only on user textual inputs but also on user query images, to improve the intentionality and diversity of search results. To tackle this task, the authors present the RL4Sugg framework, leveraging the power of Large Language Models (LLMs) with Multi-Agent Reinforcement Learning from Human Feedback to optimize the generation process. The authors have conducted extensive experiments, demonstrating the effectiveness of the RL4Sugg framework in generation-based and retrieval-based applications, and the ablation study indicates the effectiveness of RLHF.

However, I think some technical details are not well justified, for example: (1) why choosing Image-Suggestion Alignment (ISA), Image-Suggestion Generation (ISG), and Image-Suggestion Matching (ISM) as the multi-task learning for the RewardNet. Do we have ablation study results on this? (2) Why do we need Agent-D? (3) the design of Agent-D.

The innovation of this project is limited if we just look at Agent-I. It's like the multi-modal version of LLM RLHF.

**Questions:**

Could you provide more context on what the cold-start issue is for the task? The authors mention recommending suggestions when the learned knowledge is insufficient for online user queries, but it would be better to give some examples.

When do we need two agents, Agent-I and Agent-D, for this task? Why not modify Agent-I so that it can handle both the generation and diversity tasks simultaneously?

I don't feel the design of Agent-D is well-justified. In Table 4's ablation study, we have two benchmarks: (1) one without Agent-D, and (2) one replacing Agent-D with a greedy method. But what about some rule-based filtering method using query embeddings or N-grams? I think it could perform better than the greedy one.

**Reviewer Confidence:**

4: The reviewer is certain that the evaluation is correct and very familiar with the relevant literature

**Scope:**

3: The work is somewhat relevant to the Web and to the track, and is of narrow interest to a sub-community

---

### Official Review · Reviewer_Bp56 · 2023-11-29

**Novelty:** 6
**Technical Quality:** 6

**Review:**

The paper presents a  novel query formulation, called Multimodal Query Suggestion. This article has a well-organized structure.
The proposed method addresses the gap between multimodal data and query suggestions in search engines. The authors conduct extensive experiments on two real-world datasets and demonstrate the effectiveness of the proposed framework. The framework has the potential to improve the user search experience and enhance the quality of Query Suggestion systems.

The drawback of this article is that the experiments were only conducted on non-public commercial datasets, and the imagenet dataset is not a query suggestion dataset.

**Questions:**

1. Is there a publicly available multimodal query suggestion dataset？
2. Consider comparing the input image generation description with traditional query suggestion methods to further validate the effectiveness of the model.

**Reviewer Confidence:**

4: The reviewer is certain that the evaluation is correct and very familiar with the relevant literature

**Scope:**

4: The work is relevant to the Web and to the track, and is of broad interest to the community

---

### Official Review · Reviewer_oAaG · 2023-12-01

**Novelty:** 5
**Technical Quality:** 5

**Review:**

This paper propose a multi-agent based reinforcement learning from human feedback model for the multimodal query suggestion task. It is a interesting new scenario as GPT-4V provides users to upload pictures and ask questions about it.

Here are some strengths of this paper:
1. This paper proposes the task of multimodal query suggestion and designs a corresponding solution based on the technology of LLM  agents.
2. The overall introduction of the paper is clear, and the proposed method is easy to follow and understand.
3. The authors constructed and conducted a series of verification experiments to validate the effectiveness of the method. GPT-4 combined with human annotation was used on two datasets to support the evaluation.

However, there are also some weaknesses:
1. The data generated by GPT-4 is key to data collection and model training, and usually, prompt information is provided because it affects the results. However, this seems not to be reflected in the paper.
2. The method proposed in the paper is based on the OPT structure. In Tables 2 and 3, the direct performance of OPT should be compared to validate whether the improvements come from the proposed method. Additionally, significance testing should be added.
3. In the part of experimental validation, the overall setup is not clearly introduced. From the paper, it is not clear whether each model generates one suggestion or multiple suggestions.

**Questions:**

1. What is the format of the prompt for generating data? Were other models or prompt settings tried?
2. What would be the result of using the OPT model directly? Is the improvement significant?
3. In the generation task, are the evaluations based on one or multiple suggestions generated by each model?

**Ethics Review Description:**

No ethical issues.

**Reviewer Confidence:**

3: The reviewer is confident but not certain that the evaluation is correct

**Scope:**

4: The work is relevant to the Web and to the track, and is of broad interest to the community

---

### Official Review · Reviewer_n8s4 · 2023-12-02

**Novelty:** 7
**Technical Quality:** 6

**Review:**

This paper addresses the evolving landscape of information retrieval, emphasizing the need for personalized and relevant search results. While existing query suggestion systems focus primarily on textual inputs, limiting the search experience for images, this paper introduces a novel Multimodal Query Suggestion (MMQS) task. The MMQS aims to enhance the intentionality and diversity of search results by generating query suggestions based on user query images. The proposed RL4Sugg framework leverages Large Language Models (LLMs) with Multi-Agent Reinforcement Learning from Human Feedback to optimize the generation process. Through comprehensive experiments, RL4Sugg demonstrates an 18% improvement over the best existing approach. Additionally, the MMQS has been implemented in real-world search engine products, resulting in enhanced user engagement. This research contributes to the advancement of query suggestion systems and offers a fresh perspective on multimodal information retrieval.

This work is timely and relevant, addressing current challenges and concerns in the field. The literature is reviewed.

The work/approach is effectively motivated, providing clear rationale and justification for its relevance and importance.

The design of MMQS is strategically centered on two crucial properties: intentionality and diversity.

The primary objective of RL4Sugg is to generate query suggestions based on input images using the mentioned framework.  The RL4Sugg framework leverages the capabilities of Large Language Models (LLMs) with Multi-Agent Reinforcement Learning.

This work effectively tackles data generation challenges by utilizing the current capabilities of GPT language generation. It automates the collection of image-suggestion pairs and user intent annotations based on potential clicks.

The solution proposed is based on multi-agent reinforcement learning, employing two distinct agents within the framework: Agent-I, responsible for intentionality, and Agent-D, responsible for diversity. In this approach, Agent-I takes the lead by generating a set of intentional candidate suggestions. This process involves the utilization of a RewardNet and a PolicyNet, both tailored specifically for this task.

The authors evaluate RL4Sugg in terms of the generation task and the retrieval task.

**Questions:**

"For quality control, we randomly pick 10% labeled image-suggestion pairs, ask 5 other checkers to label these suggestions independently. ." -> who are these checkers and why are they appropriate? what instructions are given to them?

 " For the generation task, we let the human labelers to ..." However, it's essential to note that relying solely on human labelers introduces inherent subjectivity and potential biases into the evaluation process? how do you justify or articulate that subjectivity and biases are under conytrol?

"We observe that the DCG of RL4Sugg outperforms all other baseline models and shows good transferability." -> how is the transferability shown?

**Reviewer Confidence:**

3: The reviewer is confident but not certain that the evaluation is correct

**Scope:**

4: The work is relevant to the Web and to the track, and is of broad interest to the community

---

### Decision · Program_Chairs · 2024-01-22

**Decision:**

Accept

**Comment:**

Most reviewers agree that the paper is well-motivated and justified, the proposed method is solid and easy to follow, and the experiments are extensive. The multi-agent-based RL4Sugg model is also well-received by reviewers. Reviewer 6GYT raised concerns about the motivation of some design choices, for example, why two agents are used. If the paper gets a chance to be accepted, please carefully check the suggestions given by the reviewers, and incorporate them in the final version.